# A Novel, Tumor-Induced Osteoclastogenesis Pathway Insensitive to Denosumab but Interfered by Cannabidiol

**DOI:** 10.3390/ijms20246211

**Published:** 2019-12-09

**Authors:** Maiko Tsuchiya, Kou Kayamori, Akane Wada, Motohiro Komaki, Yae Ohata, Miwako Hamagaki, Kei Sakamoto, Tohru Ikeda

**Affiliations:** 1Department of Oral Pathology, Graduate School of Medical and Dental Sciences, Tokyo Medical and Dental University, 1-5-45, Yushima, Bunkyo-ku, Tokyo 113-8549, Japan; maiko-t.mpa@tmd.ac.jp (M.T.); kayamori.mpa@tmd.ac.jp (K.K.); miwako.mpa@tmd.ac.jp (M.H.); s-kei.mpa@tmd.ac.jp (K.S.); 2Department of Oral Diagnostic Pathology, Graduate School of Medical and Dental Sciences, Tokyo Medical and Dental University, 1-5-45, Yushima, Bunkyo-ku, Tokyo 113-8549, Japan; wadampa@tmd.ac.jp (A.W.); yae.ohata@imbim.uu.se (Y.O.); 3Department of Highly Advanced Stomatology (Periodontology), Graduate School of Dentistry, Kanagawa Dental University, 3-31-6 Tsuruya-cho, Kanagawa-ku, Yokosuka-city, Kanagawa 221-0835, Japan; m.komaki@kdu.ac.jp

**Keywords:** osteoclastogenesis, osteoclast precursor cells, tumor cells, exosomes, cannabidiol

## Abstract

Bone metabolism is strictly regulated, and impaired regulation caused by hormonal imbalances induces systemic bone loss. Local bone loss caused by tumor invasion into bone is suggested to be induced by the generation of cytokines, which affect bone metabolism, by tumor cells. The major cause of systemic and local bone losses is excess bone resorption by osteoclasts, which differentiate from macrophages by receptor activator of nuclear factor kappa-B ligand (RANKL) or tumor necrosis factor-alpha (TNF-α). We previously found a novel pathway for tumor-induced osteoclastogenesis targeting osteoclast precursor cells (OPCs). Tumor-induced osteoclastogenesis was resistant to RANKL and TNF-α inhibitors. In the present study, we confirmed that exosomes derived from oral squamous cell carcinoma (OSCC) cells induced osteoclasts from OPCs. We also showed that the depletion of exosomes from culture supernatants of OSCC cells partially interfered with osteoclastogenesis, and cannabidiol, an innoxious cannabinoid without psychotropic effects, almost completely suppressed tumor-induced osteoclastogenesis. Osteoclastogenesis and its interference by cannabidiol were independent of the expression of nuclear factor of T cell c1 (NFATc1). These results show that osteoclastogenesis induced by OSCC cells targeting OPCs is a novel osteoclastogenic pathway independent of NFATc1 expression that is partially caused by tumor-derived exosomes and suppressed by cannabidiol.

## 1. Introduction

Reductions in bone volume induce osteopenia, which may lead to osteoporosis. Osteopenia is caused by an imbalance in bone metabolism, and postmenopausal osteopenia/osteoporosis is attributed to excess bone resorption due to an estrogen deficiency [1]. Conversely, osteopenia caused by immobilization or zero gravity is considered to be strongly associated with reductions in osteogenic activity by osteoblasts [2]. Under pathological conditions, inflammatory bone resorption is associated with many types of cytokines, including tumor necrosis factor-alpha (TNF-α), interleukin 1 (IL-1), and interleukin 6 (IL-6). IL-1 and IL-6 both stimulate the expression of the major osteoclastogenic factor, receptor activator of nuclear factor kappa-B ligand (RANKL). TNF-α may induce osteoclasts and is regarded as another osteoclastogenic factor [3,4], but anti-bone resorptive effects of TNF-α have been considered through control of inflammation, and anti-TNF-α agents do not have any beneficial effect on interference of bone resorption [5]. Another major cause of pathological bone resorption occurs by metastasis or the direct invasion of malignant tumors into bone [6]. Various tumor-induced factors have been shown to stimulate osteoclastogenesis, including inflammatory cytokines, such as TNF-α, as described above. Besides inflammatory cytokines, parathyroid hormone-related peptide (PTHrP) is a well-known tumor-associated inducer of osteoclasts that stimulates the expression of RANKL [7].

Previous studies confirmed that osteoclasts were generated from macrophages by RANKL, which up-regulated the expression of the master transcriptional factor for osteoclastogenesis, nuclear factor of T cell c1 (NFATc1) [8]. TNF-α also induces osteoclasts by up-regulating the expression of NFATc1 under some pathological conditions. Based on the biological mechanisms of osteoclastogenesis, molecularly targeted drugs have been applied as therapeutic agents. Denosumab, an anti-RANKL antibody agent, is widely applicable to osteopenia and osteoporosis induced by many causes including postmenopause estrogen deficiencies and tumor-induced bone resorption. It effectively suppresses bone destruction caused by cancer metastasis including breast, prostatic, and colon cancers [9,10]. Therefore, denosumab may be used to treat giant cell tumors in bone, synonymized as osteoclastoma. Infliximab, an anti-TNF-α antibody agent, is used to mitigate bone destruction caused by rheumatic arthritis in the field of inflammatory skeletal disorders [11].

Independent of the development of these molecularly targeted agents, bisphosphonates have already been applied to patients with systemic and local bone loss caused by postmenopausal osteoporosis and tumor-associated bone resorption, particularly metastatic breast cancer and multiple myeloma [12]. Bisphosphonates are a type of phosphorous composed of a -P-C-P- chemical structure. They exhibit potent affinity to bone minerals and induce apoptosis in osteoclasts, which resorb bone containing bisphosphonates. Third-generation bisphosphonates accompanied by an amino residue are widely used to treat patients with excess bone resorption. Irrespective of their different pharmaceutical mechanisms, denosumab and amino bisphosphonates are both widely applied to the majority of disorders accompanied by excess osteoclastic bone resorption [13].

During the clinical application of bisphosphonates to patients, some side effects have been reported, and bisphosphonate-related osteonecrosis of the jaw (BRONJ), a morbidity of persistent and frequently uncontrollable osteomyelitis of the jaw bones, has been identified as the most serious side effect of amino bisphosphonates [14]. BRONJ mostly affects mandibular bone by largely unknown mechanisms. Many studies accumulated data that led to the hypothesis of BRONJ being induced by the inhibitory effects of amino bisphosphonates on wound healing and vasculogenesis. However, the mechanisms responsible for BRONJ were suggested to be more complex because denosumab and infliximab also induced osteonecrosis of the jaw [15,16]. Hence, medication-related osteonecrosis of the jaw (MRONJ) rather than BRONJ has been used to describe disorders of the jaw bone, and it may be caused by inert bone metabolism induced by anti-resorptive agents of bone. Despite the relatively low frequency of MRONJ, its intractable nature revealed the importance of elucidating the underlying mechanisms and preventing the disorder. It is also largely unclear why jaw bones, particularly the mandibular bone, appear to be specifically affected.

We previously investigated the mechanisms underlying tumor-induced osteoclastogenesis using cell lines of oral squamous cell carcinoma (OSCC), which frequently invades the jaw bones, with the aim of developing novel therapeutic agents that preferentially interfere with tumor-induced bone resorption without affecting physiological osteoclastogenesis [17]. In that study, OSCC cells induced osteoclasts from osteoclast precursor cells (OPCs), which themselves do not differentiate into osteoclasts without stimulation by RANKL or TNF-α [18]. Notably, tumor-induced osteoclastogenesis targeting OPCs was resistant to denosumab, infliximab, and osteoprotegerin (OPG) in contrast to positive controls, in which OPCs were also treated with RANKL or TNF-α. We also demonstrated that tumor-induced osteoclastogenesis occurred without the significant up-regulation of NFATc1. Based on these results, we suggest the presence of a novel pathway of osteoclastogenesis targeting OPCs that is not affected by anti-RANKL or anti-TNF-α agents [17]. In the present study, we examined the mechanisms underlying tumor-induced osteoclastogenesis targeting OPCs and found that it was partly caused by exosomes secreted by OSCC cells. We also confirmed that tumor-induced osteoclastogenesis was suppressed by cannabidiol (CBD), an innoxious cannabinoid without psychotropic effects.

## 2. Results

### 2.1. Exosomes Secreted by OSCC Cells Exhibit Osteoclastogenic Activity

Many osteoclasts were induced on the bone surface near the invasive front of OSCC but not in the stromal tissue. Based on histopathological findings, we speculated that the targets for osteoclastogenesis induced by OSCC cells were not macrophages but OPCs, which were already exposed to RANKL near the bone surface. Thus, we initially cocultured human OSCC cells and OPCs using three human OSCC cell lines, 3A, NEM, and HO-1-N1. Using the coculture system, we confirmed that human OSCC cell lines 3A and NEM induced osteoclasts from OPCs, whereas HO-1-N1 did not. OPCs did not differentiate into osteoclasts by themselves (Figure 1A,B).

We then examined the influence of exosomes secreted by OSCC cells on osteoclastogenesis. After the isolation of exosomes by ultracentrifugation of the culture supernatants of these cell lines, the amount of isolated exosomes was evaluated by the protein concentration of each suspension of exosomes. The total amount of protein in suspensions of exosomes derived from HO-1-N1 cells was significantly higher than those from 3A and NEM cells (Figure 1C). Isolated exosomes were found to have typical cup-shaped vesicular structures in observations with a transmission electron microscope (TEM) (Figure 1D), and Western blotting revealed that isolated exosomes were positive for an antibody against CD63, one of the markers of exosomes with various intensities (Figure 1E). We initially confirmed the uptake of exosomes into OPCs (Figure 1F). We then applied these exosomes to OPCs to assess their osteoclastogenic activity. Consistent with osteoclastogenesis induced by the coculture system, exosomes isolated from 3A and NEM cells, but not HO-1-N1 cells, induced osteoclasts from OPCs (Figure 2A,B). The dose dependency of osteoclastogenesis induced by exosomes was analyzed using exosomes derived from 3A cells. Osteoclastogenesis appeared at a concentration of 5 μg/mL, and the number of osteoclasts increased in a dose-dependent manner (Figure 2C,D). We then compared osteoclastogenesis in the culture supernatant of 3A cells with or without ultracentrifugation. OPCs were cultured with 40% of the culture supernatant of 3A cells as conditioned medium, and osteoclasts were generated after four days of culture (Figure 2E). When OPCs were cultured with the culture supernatant of 3A cells, excluding exosomes after ultracentrifugation in the process of exosome isolation, the number of osteoclasts was significantly less than that of osteoclasts cultured with the culture supernatant without ultracentrifugation. The number of osteoclasts was partially restored when OPCs were cultured with the ultracentrifuged culture supernatant mixed with the exosomal pellet (Figure 2F).

### 2.2. Osteoclastogenesis Induced by OSCC Cells was Resistant to Denosumab but was Effectively Suppressed by CBD and Dimethyl Amiloride.

The present results strongly suggested that osteoclastogenesis induced by OSCC cells was at least partly caused by exosomes. We then investigated whether agents that inhibit the secretion of exosomes affected tumor-induced osteoclastogenesis. We selected two drugs, CBD and dimethyl amiloride (DMA) as candidates, both of which have been shown to suppress the secretion of exosomes [19,20]. In contrast to the ineffectiveness of the administration of denosumab, CBD and DMA both significantly inhibited osteoclastogenesis induced by 3A cells. Conversely, denosumab completely inhibited RANKL-induced osteoclastogenesis, while neither CBD nor DMA affected osteoclastogenesis induced by RANKL (Figure 3A,B). The inhibitory effects of CBD on osteoclastogenesis induced by 3A cells were more potent than those of DMA at various concentrations of each drug. At a concentration higher than 4 μM, CBD almost completely inhibited osteoclastogenesis (Appendix A). DMA significantly inhibited osteoclastogenesis induced by 3A cells at a concentration higher than 15 nM (Appendix A). In contrast, neither CBD nor DMA affected osteoclastogenesis induced by RANKL up to high concentrations (Appendix A). To examine the inhibited secretion of exosomes by CBD or DMA in more detail, we quantified the number of exosomes with or without the administration of CBD or DMA using the q-Nano System. CBD did not inhibit the secretion of exosomes by 3A cells (Figure 3C), whereas DMA slightly suppressed the secretion of exosomes and microvesicles (MVs) by 3A cells (Figure 3D). These results suggest that the mechanism underlying the inhibition of osteoclastogenesis by CBD differs from that by DMA, and they also indicate that exosomes of 3A cells treated with CBD included some inhibitory factor(s) of tumor-induced osteoclastogenesis. Therefore, we isolated exosomes from the culture supernatant of 3A cells with or without the CBD treatment and applied them to a coculture of OPCs and 3A cells. As shown in Figure 3E,F, exosomes isolated from culture supernatants of 3A cells with or without the CBD treatment significantly increased osteoclastogenesis induced by the coculture, and, consequently, exosomes isolated from 3A cells treated with CBD did not include inhibitory factors of osteoclastogenesis induced by the coculture of OPCs and 3A cells. We also examined osteoclastogenesis induced by not only exosomes, defined as membrane particles ranging between approximately 50 and 150 nm in diameter, but also by MVs, defined as membrane particles larger than exosomes. Osteoclastogenesis induced by MVs isolated from the culture supernatant of 3A cells was similar to that induced by exosomes (Figure 3G,H).

### 2.3. Osteoclastogenesis Induced by OSCC Cells was not Dependent on the Up-Regulation of NFATc1

The present results strongly suggested that osteoclastogenesis induced by OSCC cells targeting OPCs occurred via a unique pathway that was independent of RANKL. Therefore, we investigated whether osteoclastogenesis induced by OSCC cells depended on the up-regulation of the master transcriptional factor of osteoclastogenesis, NFATc1, using a quantitative PCR analysis. We confirmed that osteoclastogenesis from OPCs treated with RANKL was accompanied by the up-regulated expression of NFATc1, while the administration of denosumab inhibited the expression of NFATc1. Conversely, the administration of CBD, which did not affect osteoclastogenesis from OPCs treated with RANKL, did not significantly alter the expression of NFATc1 (Figure 4). These results showed that the number of osteoclasts induced by RANKL from OPCs correlated with the expression level of NFATc1.

In contrast to RANKL-induced osteoclastogenesis, the expression of NFATc1 remained low in a coculture of OPCs and 3A cells, irrespective of the generation of osteoclasts. Unexpectedly, the administration of denosumab, which did not affect osteoclastogenesis in the coculture of OPCs and 3A cells, significantly up-regulated the expression of NFATc1 but did not alter the low-level expression of NFATc1. Notably, the administration of CBD did not inhibit the expression of NFATc1, irrespective of the significant reduction in the number of osteoclasts. These results showed that osteoclastogenesis induced by OSCC cells from OPCs did not correlate with the expression of NFATc1.

## 3. Discussion

We previously reported that OSCC cells induced osteoclasts from OPCs, which themselves do not have the capacity to differentiate into osteoclasts without further stimulation by RANKL or TNF-α [17]. The concept of OPCs was initially proposed by Mizoguchi et al., and they defined OPCs as macrophages treated with RANKL for 24 hours that had neither growth activity nor the capacity to differentiate into osteoclasts without a further stimulation with RANKL. They indicated that OPCs were widely distributed throughout the body through the bloodstream, and they speculated that they played a pivotal role in bone resorption [18]. We also showed that osteoclastogenesis was resistant to osteoprotegerin, denosumab, or infliximab. Microarray analysis of 3A and HO-1-N1 cells revealed low expression of RANKL in both OSCC cells and higher expression of TNF-α in 3A cells. Although we did not analyze the concentration of RANKL in the culture supernatant, we suggest that osteoclastogenesis of 3A cells was not caused by up-regulation of RANKL nor higher expression of TNF-α in 3A cells. Furthermore, we suggest that osteoclastogenesis was independent of the up-regulation of NFATc1 [17]. However, the mechanisms underlying tumor-induced osteoclastogenesis targeting OPCs have not yet been elucidated in detail.

In the present study, we demonstrated for the first time that exosomes and MVs derived from cancer cells had the ability to generate osteoclasts from OPCs. Exosomes have been shown to influence the biological behavior of tumor cells, including growth, drug resistance, angiogenesis, invasion, and metastasis [21,22]. In addition to these pivotal biological behaviors of tumor cells, a wide variety of functions of exosomes secreted by tumor cells have been reported [23,24]. Exosomes isolated from the culture supernatant of OSCC cells that exhibited osteoclastogenic activity in coculture with OPCs induced osteoclasts, whereas those isolated from the supernatant of OSCC cells that did not exhibit osteoclastogenic activity in coculture with OPCs did not. This relationship between the osteoclastogenic activity of OSCC cells in the coculture system and that of exosomes isolated from the culture supernatant of OSCC cells strongly suggests that exosomes secreted by OSCC cells participate in osteoclastogenesis induced by OSCC cells in the coculture system. Some soluble factors generated by OSCC cells induce osteoclasts based on histopathological findings showing that invading OSCC cells do not directly make contact with bone surfaces, but they are intercalated in stromal cells. The present results suggested that exosomes secreted by OSCC cells were included in OPCs near the bone surface, which made contact with bone marrow stromal cells near bone tissue expressing low levels of RANKL [4]. Previous studies suggested that tumor-derived exosomes affect bone metabolism [25,26,27]. Many studies indicated that the various biological functions of exosomes were expressed by miRNAs included in exosomes, which were generally considered to be transmitters of functional miRNAs [28,29,30]. Recently, miRNAs in exosomes secreted by lung cancer cells and breast cancer cells were reported to promote RANKL-induced osteoclastogenesis [25,26]. Analysis of miRNAs in exosomes secreted by OSCC cells and their association with osteoclastogenesis targeting OPCs is needed.

Collectively, these findings and the present results indicate that certain cancer cells have the potential to invade bone through the secretion of exosomes, and further studies using other cancer cells are needed to obtain a clearer understanding of this novel mechanism of tumor-induced osteoclastogenesis.

Another important point of the present study is that osteoclastogenesis induced by the coculture of OPCs and OSCC cells was inhibited by CBD irrespective of increases in the number of exosomes secreted by tumor cells. We also demonstrated that the osteoclastogenic activity of the culture supernatant of 3A cells, in which exosomes were removed by ultracentrifugation, was significantly decreased, but not completely absent, and restored by the addition of exosomes that were removed by ultracentrifugation. These results indicate that CBD does not interfere in the secretion of exosomes from 3A cells, but it inhibits osteoclastogenesis induced by exosomes. In addition, these results also strongly suggest that OSCC cells induce osteoclasts by multiple mechanisms involving exosomes and other unknown mechanisms, both of which were affected by CBD but not by denosumab. To confirm our hypothesis, the other mechanisms underlying the inhibition of osteoclastogenesis by CBD need to be elucidated.

Accumulated evidence shows that endocannabinoids composed of anandamide and 2-arachidonoylglycerol (2-AG), both of which are generated endogenously, regulate bone metabolism [31]. There are two specific cannabinoid receptors, cannabinoid receptor type one (CB1) and type two (CB2) [32]. Cannabinoids are also known to interact with other types of receptors, and transient receptor potential vanilloid type 1 (TRPV1) is involved in authentic functional receptors to express the biological activities of endocannabinoids [33]. In addition to these endocannabinoids, many types of cannabinoids have also been shown to express their biological activities through these three receptors [32]. In the field of bone cell biology, these three receptors were found to be expressed in both osteoblasts and osteoclasts. The stimulation of CB1 and TRPV1 with their agonists has been suggested to increase bone resorption [34]. Conversely, a previous study indicated that the stimulation of CB2 with the agonist prevented bone resorption [31]. The distinct roles of CB1 and CB2 in bone metabolism raise the hypothesis that the balance between CB1 and CB2 regulates bone metabolism, and the regulation of endocannabinoids and expression of the receptors CB1, CB2, and TRPV1 may contribute to the regulation of bone metabolism [33].

CBD has low affinity to CB1 and CB2. In addition, although CBD was reported to be a TRPV1 agonist, biological activity in bone metabolism has not yet been demonstrated [32]. The orphan G protein receptor GPR55 was also shown to act as one of the receptors for some cannabinoids, and CBD functions as an antagonist of GPR55 [35]. Whyte et al. reported that GPR55 affected osteoclastogenesis and osteoclast functions [36]. In their study, the administration of CBD increased osteoclastogenesis induced by RANKL, but it suppressed osteoclast activity. Conversely, the GPR55 agonist O-1602 suppressed RANKL-induced osteoclastogenesis, but it stimulated osteoclast activity. Animal experiments revealed that the administration of CBD to mice suppressed bone resorption. They also showed that GPR55-deficient mice had a greater bone volume than wild-type mice. These results suggest that the GPR55 pathway contributed to bone resorption, and CBD, one of the antagonists of GPR55, exerted suppressive effects on bone resorption.

The present results clearly showed that CBD inhibited osteoclastogenesis induced by OSCC cells, but it did not affect that induced by RANKL. These discrepancies may be caused by differences in the targeting cells. In the present study, we used OPCs for in vitro cultures. We previously demonstrated that osteoclastogenesis induced by OSCC cells targeting OPCs was resistant to denosumab, infliximab, and OPG, and we also suggested that the mechanism underlying osteoclastogenesis was independent of the up-regulation of NFATc1 [17]. In the present study, we confirmed that osteoclastogenesis induced by OSCC cells was not accompanied by the up-regulation of NFATc1, and the administration of CBD or DMA did not affect the expression of NFATc1 irrespective of their inhibition of osteoclastogenesis. We also demonstrated that the administration of denosumab in the coculture of OPCs and 3A cells significantly up-regulated the expression of NFATc1. The biological significance of the up-regulation is not clear, and we should verify the up-regulation at different concentrations of denosumab and carefully evaluate the unexpected phenomenon.

Collectively, the present results strongly suggest that the biological mechanism underlying osteoclastogenesis targeting OPCs is independent of that targeting non-treated macrophages, and we offer the novel concept that the regulation of bone metabolism targeting OPCs is a new therapeutic target for bone diseases, particularly tumor-induced bone resorption. Furthermore, because of the risk of the serious side effect, MRONJ in patients treated with amino bisphosphonates or denosumab, a clearer understanding of the mechanisms underlying osteoclastogenesis targeting OPCs will be of significant value for the development of new therapeutic agents that preferentially act on specific types of pathological bone resorption without affecting physiological bone metabolism. Further studies on exosome-induced osteoclastogenesis and the CBD-sensitive pathway of osteoclastogenesis targeting OPCs are essential.

## 4. Materials and Methods

### 4.1. Cell Culture

Three human OSCC cell lines, 3A, NEM, and HO-1-N1, were used. These cell lines were maintained in high-glucose Dulbecco’s Modified Eagle’s Medium (DMEM) (Sigma-Aldrich, St. Louis, MO, USA) containing 10% fetal bovine serum (FBS) (Sigma-Aldrich) and 1% penicillin–streptomycin (Nacalai Tesque Inc., Kyoto, Japan) as described previously [4,16]. The 3A and NEM cells lines were kindly provided by Professor Emeritus Nobuo Tsuchida, Tokyo Medical and Dental University. The HO-1-N1 cell line was purchased from the Japanese Collection of Research Bioresources (Osaka, Japan).

### 4.2. Preparation of Mouse Bone Marrow Macrophages and OPCs

Mouse bone marrow macrophages were prepared as described previously [16]. Bone marrow cells were obtained from the tibiae and femurs of five-week-old female DDY mice (Sankyo Lab service, Shizuoka, Japan) by flushing with alpha minimum essential medium (α-MEM, Sigma-Aldrich) using a 23-gauge needle. After dissociation by pipetting, bone marrow cells were cultured overnight in α-MEM containing 10% FBS (Sigma-Aldrich) and 1% penicillin–streptomycin (Nacalai Tesque) further supplemented with 50 ng/mL macrophage colony-stimulating factor (M-CSF) (Leucoprol, Kirin-Kyowa Hakko Co., Osaka, Japan). Uncontacted cells were collected and centrifuged with Lymphoprep^TM^ (Axis Shield PoC AS, Oslo, Norway) following the manufacturer’s instructions to isolate macrophages. Isolated macrophages were expanded by culturing in medium for four days. OPCs were generated from expanded macrophages by culturing in medium further supplemented with 100 ng/mL of recombinant human RANKL (PeproTech, Rocky Hill, NJ, USA) for 24 hours as described previously [17,18]. Animal experiments were performed following the Guidelines for Animal Experimentation of Tokyo Medical and Dental University with the official approval of the committee (Approval no. A2017-039A).

### 4.3. In Vitro Osteoclastogenesis Induced by OSCC Cells or RANKL

In the coculture system, 3 × 10^4^ OPCs and 3 × 10^3^ cancer cells were cocultured with α-MEM medium in each well of a 48-well plate to generate tumor-induced osteoclasts. Osteoclastogenesis was evaluated on day four of the coculture after staining for tartrate-resistant acid phosphatase activity, as reported previously [37]. RANKL-induced osteoclasts were generated by culturing 3 × 10^4^ OPCs in α-MEM medium further supplemented with 50 ng/mL of M-CSF and 100 ng/mL of recombinant human RANKL (Peprotech) for four days. M-CSF was added every day to the culture. Each medium was changed after two days of culture. TRAP-positive cells with three or more nuclei were considered to be osteoclasts.

The effects of inhibitory agents on osteoclastogenesis were evaluated as follows. In cocultures of OPCs and 3A cells or OPCs cultured in medium supplemented with RANKL, 100 μg/mL of denosumab (Ranmark; Daiichi-Sankyo Co. Ltd., Tokyo, Japan) was added, cultured for four days, and the number of osteoclasts was quantitated after TRAP staining. To examine the effects of CBD or DMA on osteoclastogenesis, 1 or 5 μM of CBD (C6395, Sigma-Aldrich) or 1 or 15 nM of DMA (5-(N,N-dimethyl)-Amiloride; Cayman Chemical, Ann Arbor, MI, USA) was added to cocultures of OPCs and 3A cells or OPCs cultured in medium supplemented with RANKL. Each medium was changed after two days of culture, and osteoclastogenesis was evaluated on day four of the culture, as described above. The optimum concentration of CBD and DMA was evaluated by the application of different concentrations of CBD or DMA to the cocultures, as described previously [19,38].

### 4.4. Preparation of Exosomes and MVs

OSCC cells (3A, NEM, and HO-1-N1) were plated in ten 82-mm culture dishes and cultured with DMEM growth medium. When cells reached 80%–90% confluency, the medium was changed to serum-free DMEM. After 48 hours, 100 mL of the culture supernatant was collected, centrifuged at 2000× *g* at 4 °C for 20 mins, and filtered using a filter unit with a pore size of 0.2 μm (Kurabo, Osaka, Japan) to remove cellular debris. The filtered culture supernatant was centrifuged at 150,000× *g* at 4 °C for 70 mins (Optima XE-90 ultracentrifuge with a swing rotor, SW41Ti; Beckman Coulter, Inc., Brea, CA, USA). The pellets were suspended in PBS, re-centrifuged, and suspended in 1 mL of PBS [39,40].

Ten microliters of the exosome suspension was dropped onto a copper grid with a carbon-support film, washed with water, and treated with uranium acetate. The shapes of the prepared exosomes were visualized using a transmission electron microscope (TEM) (H-7100, Hitachi Ltd., Tokyo, Japan). To evaluate the quantity of exosomes, the protein concentration of each exosome suspension was measured using a Micro BCA^TM^ Protein Assay Kit (Thermo Fisher Scientific, MA, USA). Exosomes were also detected by Western blotting using an anti-CD63 antibody (MX-49.129.5, Santa Cruz Biotechnology, Santa Cruz, CA). MVs in culture supernatants were prepared from the supernatant of 3A cells by centrifugation at 300× *g* at 4 °C for five minutes to remove cell debris, and this was followed by centrifugation at 16,500× *g* at 4 °C for 30 minutes [20,41]. The pellets were suspended in PBS. The size distribution and number of exosomes or MVs were analyzed using a qNano System (Izon Science, Christchurch, New Zealand) with nanopore NP100 to detect exosomes and with nanopore NP800 to detect MVs. The analyses were consigned to MEIWAFOSIS Co., Ltd., Tokyo, Japan.

### 4.5. Detection of the Uptake of Exosomes into OPCs

To confirm the uptake of exosomes into OPCs, exosomes were labeled using a PKH67 Green Fluorescent Cell Linker Kit (Sigma-Aldrich) according to the manufacturer’s protocol with minor modifications as previously described [42]. Three hundred microliters of each exosome suspension was mixed with 100 μL of Diluent C. Stain solution was prepared by adding 1.4 μL of PKH67 to another 300 μL of Diluent C. The exosome solution and stain solution were mixed, incubated at room temperature for four minutes, and the labeling reaction was stopped by adding 700 μL of FBS to the mixed solution. Labeled exosomes were added to OPCs and incubated at 37 °C for 24 hours. After the incubation, cells were fixed with 2% PFA solution. OPCs were immunostained using an anti-CD68 antibody (KP1, Dako, CA, USA) labeled with Alexa Fluor 594 goat anti-mouse IgG (A11005, Invitrogen) and observed using an Axioskop 2 plus fluorescent microscope (Carl Zeiss, Jena, Germany).

### 4.6. In Vitro Osteoclastogenesis Induced by Exosomes

One to 15 μg/mL of exosomes or 15 μg/mL of MVs were added to 3 × 10^4^ OPCs in each well of a 48-well plate and cultured in α-MEM containing 10% FBS (Sigma-Aldrich) and 1% penicillin–streptomycin (Nacalai Tesque) further supplemented with 50 ng/mL M-CSF (Leucoprol, Kirin-Kyowa Hakko Co.). Culture media were changed after two days of culture, and M-CSF was added every day. On day four of the culture, osteoclastogenesis was evaluated as described above.

### 4.7. Quantitative RT-PCR

OPCs and 3A cells were cocultured in the wells of a 24-well plate using two-fold the number of cells cultured in the wells of 48-well plates under the conditions described above. In some cocultures, 5 μM of CBD or 100 μg/mL of denosumab was added. After two days of culture, the medium was changed, and on day four of the culture, cells were harvested, and total RNA was extracted from cells using a NucleoSpin RNA Kit (Macherey-Nagel, Duren, Germany). Total RNA was reverse-transcribed into cDNA according to the method described previously [43]. Quantitative real-time PCR was performed using the FastStart Essential DNA Green Master Mix (Roche Applied Science, Penzburg, Germany) and Light Cycler Nano (Roche Diagnostics, Basel, Switzerland). The relative expression level of each mRNA was calculated by the comparative CT method with GAPDH as an internal control. Each experiment was repeated at least three times. Primer sequences to analyze the expression of mouse NFATc1 and mouse GAPDH in OPCs were as follows: mNFATc1, 5’- TGCTCCTCCTCCTGCTGCTC-3’ (forward) and 5’- CGTCTTCCACCTCCACGTCG -3’ (reverse) [8]; mGAPDH, 5’- CATGGCCTTCCGTGTTCCTA -3’ (forward) and 5’- GCGGCACGTCAGATCCA -3’ (reverse) [17].

### 4.8. Statistical Analysis

All results were given as the mean ± SEM of independent replicates. The difference between two groups was analyzed using the Student’s *t*-test. The difference among three or more groups was analyzed using a one-way analysis of variance and Tukey–Kramer or Dunnett’s post hoc multiple comparisons. *p* values < 0.05 were considered to be significant.

## Figures and Tables

**Figure 1 ijms-20-06211-f001:**
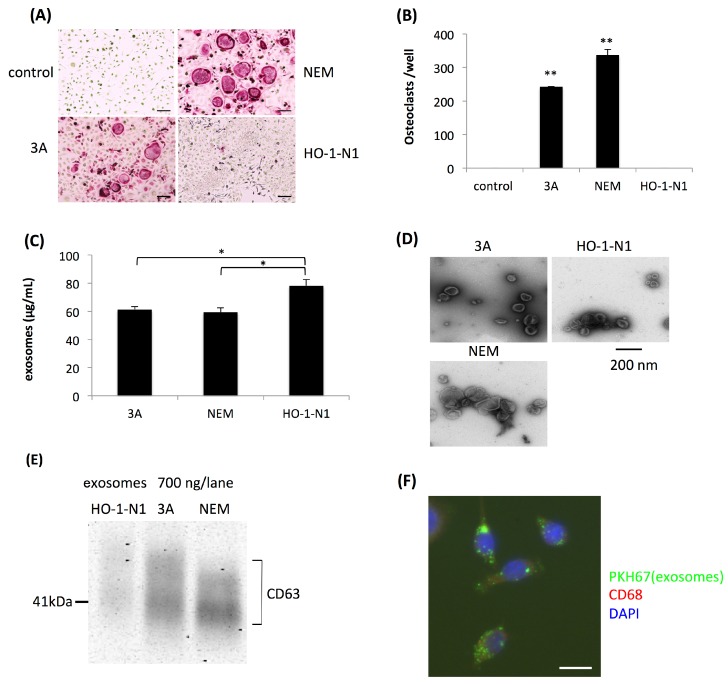
Induction of osteoclasts by oral squamous cell carcinoma (OSCC) cells from osteoclast precursor cells (OPCs) using a coculture system. (**A**) Representative views of tartrate-resistant acid phosphatase (TRAP) staining four days after the coculture of OPCs and 3A, NEM, or HO-1-N1 OSCC cells, and a negative control without a coculture. Bars represent 100 μm. (**B**) Quantitative data on the number of induced osteoclasts relative to that of the control. Values are the mean ± SEM of three wells. ** *p* < 0.001 (one-way ANOVA with the Tukey–Kramer method). (**C**) The yield of exosomes from each OSCC cell line evaluated by the colorimetry of protein concentrations. Values are the mean ± SEM of three experiments. * *p* < 0.05 (one-way ANOVA with Tukey–Kramer methods). (**D**) Morphological views of exosomes isolated from each culture supernatant of OSCC cells by TEM analysis. (**E**) Detection of CD63 in exosomes isolated from each culture supernatant of OSCC cells by Western blotting. (**F**) Incorporation of exosomes labeled with PKH-67 in OPCs 24 hours after their application. The bar represents 20 μm.

**Figure 2 ijms-20-06211-f002:**
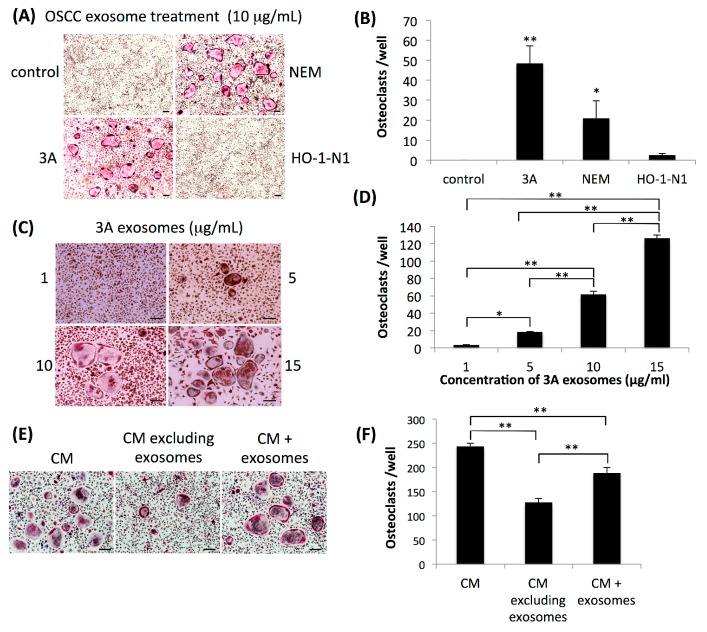
Effects of exosomes derived from OSCC cell lines on osteoclastogenesis. (**A**) Representative views of TRAP staining four days after the application of exosomes derived from 3A, NEM, or HO-1-N1 cells to OPCs. Control represents OPCs without the application of exosomes. Bars represent 100 μm. (**B**) Quantitative data on the number of induced osteoclasts relative to that of the control. Values are the mean ± SEM of more than three wells. ** *p* < 0.001, * *p* < 0.05 (one-way ANOVA with the Tukey–Kramer method). (**C**) Representative views of TRAP staining four days after the application of each concentration of exosomes derived from 3A cells to OPCs. Bars represent 100 μm. (**D**) Quantitative data on the number of induced osteoclasts relative to that of the culture application of 1 μg/mL of exosomes. Values are the mean ± SEM of three wells. ***P* < 0.001, * *p* < 0.05 (one-way ANOVA with the Tukey–Kramer method). (**E**) Representative views of TRAP staining four days after the cultivation of OPCs with conditioned medium containing 40% of the culture supernatant of 3A cells (CM), ultracentrifuged culture supernatant of 3A cells (CM excluding exosomes), or ultracentrifuged culture supernatant of 3A cells mixed with isolated exosomes (CM + exosomes). Bars represent 100 μm. (**F**) Quantitative data on the number of induced osteoclasts relative to that of the culture of OPCs with CM. Values are the mean ± SEM of three wells. ** *p* < 0.001 (one-way ANOVA with the Tukey–Kramer method).

**Figure 3 ijms-20-06211-f003:**
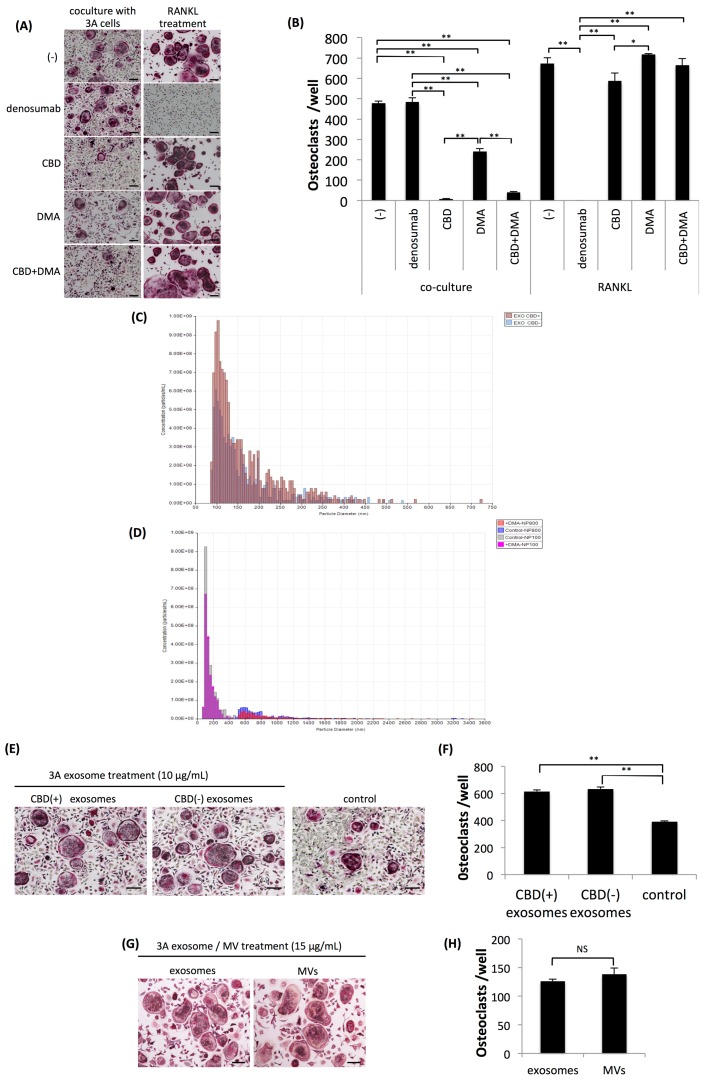
Effects of cannabidiol (CBD), dimethyl amiloride (DMA), or denosumab on osteoclastogenesis. (**A**) Representative views of TRAP staining four days after the coculture of OPCs and 3A cells (left lane) and OPCs further treated with 100 ng/mL of receptor activator of nuclear factor kappa-B ligand (RANKL) (right lane) with or without treatment with denosumab (100 μg/mL), CBD (5 μM), or DMA (15 nM). Bars represent 100 μm. (**B**) Quantitative data on the number of induced osteoclasts. Values are the mean ± SEM of three wells. ** *p* < 0.001, * *p* < 0.05 (one-way ANOVA with the Tukey–Kramer method). (**C**) Quantitation of the number of exosomes in the culture supernatants of 3A cells with (EXO CBD+) or without (EXO CBD‒) a treatment with CBD. (**D**) Quantitation of the number of exosomes in the culture supernatant of 3A cells with (+DMA NP100) or without (Control NP100) a treatment of DMA, and the number of microvesicles (MVs) with (+DMA NP800) or without (Control NP800) a treatment of DMA. (**E**) Representative views of TRAP staining four days after the coculture of OPCs and 3A cells (control), coculture of OPCs and 3A cells further supplemented with 10 μg/mL of exosomes derived from the culture supernatant of CBD-treated 3A cells (CBD (+) exosomes), supplemented with exosomes derived from the culture supernatant of 3A cells untreated with CBD (CBD (-) exosomes), and without the supplementation of exosomes (control). Bars represent 100 μm. (**F**) Quantitative data on the number of induced osteoclasts. Values are the mean ± SEM of three wells. ** *p* < 0.001 (one-way ANOVA with the Tukey–Kramer method). (**G**) Representative views of TRAP staining four days after the application of 15 μg/mL of exosomes or MVs derived from the culture supernatant of 3A cells to OPCs. Bars represent 100 μm. (**H**) Quantitative data on the number of induced osteoclasts. Values are the mean ± SEM of three wells. NS: Not significant (*t*-test).

**Figure 4 ijms-20-06211-f004:**
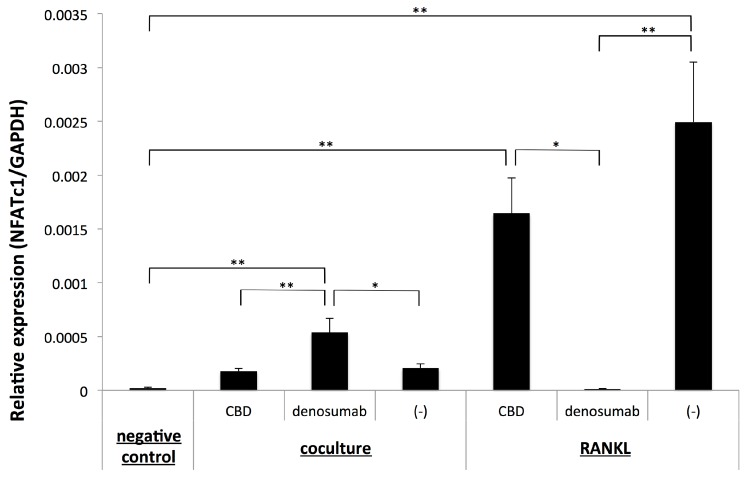
Quantitative RT-PCR analysis of the expression of NFATc1 with or without the treatment with CBD or denosumab in a coculture of OPCs and 3A cells (coculture) or OPCs further treated with 100 ng/mL of RANKL. The negative control represents OPCs neither co-cultured nor treated further with RANKL. Values are the mean ± SEM of three wells. ** *p* < 0.001, * *p* < 0.05 (one-way ANOVA with the Tukey–Kramer method).

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
