# Peer review of "A Novel, Tumor-Induced Osteoclastogenesis Pathway Insensitive to Denosumab but Interfered by Cannabidiol"

_ijms, 2019, doi:10.3390/ijms20246211_

Round 1
Reviewer 1 Report
In this manuscript entitled “A Novel Tumor Induced Osteoclastogenesis Pathway Insensitive to Denosumab but Interfered by Cannabidiol", authors in their model showed that exosomes derived from oral squamous cell carcinoma are key drivers in cancer-induced osteoclastogenesis and cannabidiol inhibited the osteoclast differentiation in NFAT1 independent fashion. The study is well designed and their experiments are clearly supporting the conclusions.
However, I have a concern about the osteoclast differentiation in cocultures:
The authors showed that Denosumab is ineffective in co-cultures. In figure 2e & 2f condition medium without exosomes from 3a cells significantly induced the osteoclastogenesis. In figure 3a there is no difference observed between co-cultures control and Denosumab treatment group. In figure 5 coculture experiments Denosumab significantly increased the expression of NFATc1. I strongly recommend the authors to reconcile their data by doing a dose-dependent study with Denosumab.Determination of RANKL secretion from cancer cells using the ELISA method provides more information about why Denosumab is not effective in co-cultures.
In figure 4a. The authors showed that the addition of exosomes significantly increased osteoclast differentiation and there is no difference observed between with and without CBD treatment. Authors should clearly discuss the fact that CBD treatment didn’t affect the secretion of exosomes from the 3A tumors cells but CBD inhibited the exosomes induced osteoclast differentiation.
I also recommend the authors to arrange the figures for better understanding of data, for instance, moving figure 4 data to figure 3. To explain CBD is not effecting the exosomes expression but inhibiting the exosomes mediated osteoclastogenesis.
Minor comments
I suggest mentioning the concentration of CBD, DMA, and Denosumab in figures and their legends. In Figure 3c & 3d, the dot plot representation of exosomes quantification offers more clarity.Author Response
Response to the Reviewer 1
Thank you very much for your valuable comments. We carefully read your comments and modified the manuscript. We believe that this revised manuscript has been improved by modifications in response to the reviewers’ comment, and we hope you will agree to consider this revised manuscript appropriate for publication in The International Journal of Molecular Sciences.
The authors showed that Denosumab is ineffective in co-cultures. In figure 2e & 2f condition medium without exosomes from 3a cells significantly induced the osteoclastogenesis. In figure 3a there is no difference observed between co-cultures control and Denosumab treatment group. In figure 5 coculture experiments Denosumab significantly increased the expression of NFATc1. I strongly recommend the authors to reconcile their data by doing a dose-dependent study with Denosumab.
In this study, we did not analyze the expression of NFATc1 at different concentration of denosumab. In the previous study [ref. 17], we showed that the expression of RANKL was low in 3A and HO-1-N1 cells and the expression of TNF-a was higher in 3A cells than in HO-1-N1 cells. Hence, considering with ineffective of high concentration of osteoprotegerin (300 ng/mL) or denosumab (100 mg/mL), both of which completely inhibited osteoclastogenesis induced by 100 ng/mL of RANKL, or infliximab (100 mg/mL), which completely inhibited osteoclastogenesis induced by 100 ng/mL of TNF-a, in osteoclastogenesis of the co-cultures, we thought that the osteoclastogenesis induced by the co-culture was caused by neither up-regulation of RANKL nor higher expression of TNF-a in 3A cells. Based on these results in the previous study, we also used the same concentration (100 mg/mL) of denosumab in this study.
As described in the discussion section (page 10, lines 14-22 in the revised version), we thought that the osteoclastogenesis induced by conditioned medium without exosomes shown in figure 2e and 2f was caused by unknown mechanisms inhibited by CBD rather than denosumab.
In figure 5, we demonstrated that denosumab significantly up-regulated the expression of NFATc1. The biological significance of the low-level up-regulation remains uncertain. As the reviewer suggested, we should verify the up-regulation at different concentration of denosumab and carefully evaluate the unexpected phenomenon.
Following the reviewer’s comments, we changed “We also showed that osteoclastogenesis was resistant to denosumab or infliximab.” (page 8, lines 22 and 23 in the original version) to “We also showed that osteoclastogenesis was resistant to osteoprotegerin, denosumab, or infliximab.” (page 9, lines 14 and 15 in the revised version), and added “Microarray analysis of 3A and HO-1-N1 cells revealed low expression of RANKL in both OSCC cells and higher expression of TNF-a in 3A cells. Although we did not analyze the concentration of RANKL in the culture supernatant, we suggest that osteoclastogenesis of 3A cells was not caused by up-regulation of RANKL nor higher expression of TNF-a in 3A cells.” (page 9, lines 15-19) and “We also demonstrated that the administration of denosumab in the coculture of OPCs and 3A cells significantly up-regulated the expression of NFATc1. The biological significance of the up-regulation is not clear, and we should verify the up-regulation at different concentrations of denosumab and carefully evaluate the unexpected phenomenon.” (page 11, lines 6-10) in the discussion section of the revised version.
Determination of RANKL secretion from cancer cells using the ELISA method provides more information about why Denosumab is not effective in co-cultures.
As pointed out by the reviewer, analysis of the concentration of RANKL in culture supernatants of OSCC cells provides additional important information. However, we did not quantify RANKL concentration in culture supernatants of OSCC cells in this study. This comment is thought to be closely associated with the initial comments. As mentioned above, we showed that the expression of RANKL was low in 3A and HO-1-N1 cells in the previous study [ref. 17], and added “Microarray analysis of 3A and HO-1-N1 cells revealed low expression of RANKL in both OSCC cells and higher expression of TNF-a in 3A cells. Although we did not analyze the concentration of RANKL in the culture supernatant, we suggest that osteoclastogenesis of 3A cells was not caused by up-regulation of RANKL nor higher expression of TNF-a in 3A cells.” in the discussion section of the revised version (page 9, lines 15-19).
In figure 4a. The authors showed that the addition of exosomes significantly increased osteoclast differentiation and there is no difference observed between with and without CBD treatment. Authors should clearly discuss the fact that CBD treatment didn’t affect the secretion of exosomes from the 3A tumors cells but CBD inhibited the exosomes induced osteoclast differentiation.
Following the reviewer’s advice, we changed “Another important point of the present study is that osteoclastogenesis induced by the coculture of OPCs and OSCC cells was inhibited by CBD irrespective of increases in the number of exosomes secreted by tumor cells. We also demonstrated that the osteoclastogenic activity of the culture supernatant of 3A cells, in which exosomes were removed by ultracentrifugation, was significantly decreased, but not completely absent. These results strongly suggest that OSCC cells induce osteoclasts by multiple mechanisms involving exosomes and other unknown mechanisms, both of which were affected by CBD.” in the discussion section of the original version (page 9, lines 18-24) to “Another important point of the present study is that osteoclastogenesis induced by the coculture of OPCs and OSCC cells was inhibited by CBD irrespective of increases in the number of exosomes secreted by tumor cells. We also demonstrated that the osteoclastogenic activity of the culture supernatant of 3A cells, in which exosomes were removed by ultracentrifugation, was significantly decreased, but not completely absent, and restored by the addition of exosomes that were removed by ultracentrifugation. These results indicate that CBD does not interfere in the secretion of exosomes from 3A cells, but it inhibits osteoclastogenesis induced by exosomes. In addition, these results also strongly suggest that OSCC cells induce osteoclasts by multiple mechanisms involving exosomes and other unknown mechanisms, both of which were affected by CBD but not by denosumab.” in the discussion section of the revised version (page 10, lines 14-22).
I also recommend the authors to arrange the figures for better understanding of data, for instance, moving figure 4 data to figure 3. To explain CBD is not effecting the exosomes expression but inhibiting the exosomes mediated osteoclastogenesis.
Following the reviewer’s suggestion, we combined Figure 3 and Figure 4 in the revised version.
Minor comments
I suggest mentioning the concentration of CBD, DMA, and Denosumab in figures and their legends. In Figure 3c & 3d, the dot plot representation of exosomes quantification offers more clarity.
As described above, we combined Figure 3 and Figure 4. Considering the layout of new Figure 3 composed of 8 figures, we added concentrations of CBD, DMA and denosumab only in the figure legend (“with or without treatment with denosumab (100 mg/mL), CBD (5 mM), or DMA (15 nM)”, page 8, line 3 in the revised version). We think that these additions are greatly helpful to better understanding of the multiple figures by the readers.
qNano analyses were consigned to MEIWAFOSIS Co., Ltd., and dot plot data was not available. Hence, we did not change the column charts. However, we enlarged Figure 3C and 3D in the revised version.
Other changes made
We corrected some typographical and grammatical errors using an English Editing Service provided by MDPI.
Reviewer 2 Report
The manuscript is interesting. The paper increases knowledgment upon basis of bone metabolism.
However, I suggest to the authors to underlyne that anti-TNF Alpha agents do not interfere with bone reabsorption adding as reference (see Murdaca et al papers upon TNF alpha inhibitors) and underlyne in the introduction this topic .
Author Response
Response to the Reviewer 2
Thank you very much for your valuable comments. We carefully read your comments and modified the manuscript. We believe that this revised manuscript has been improved by modifications in response to the reviewers’ comment, and we hope you will agree to consider this revised manuscript appropriate for publication in The International Journal of Molecular Sciences.
The manuscript is interesting. The paper increases knowledgment upon basis of bone metabolism.
However, I suggest to the authors to underlyne that anti-TNF Alpha agents do not interfere with bone reabsorption adding as reference (see Murdaca et al papers upon TNF alpha inhibitors) and underlyne in the introduction this topic.
Following the reviewer’s advice, we added a new reference (Kawai, V.K.; Stein, C.M.; Perrien, D.S.; Griffin, M.R. Effects of anti-tumor necrosis factor alpha agents on bone. Curr Opin Rheumatol 2012, 24, 576-585, doi:10.1097/BOR.0b013e328356d212.), which clearly described that anti-TNF-a agents were ineffective to bone resorption as reference [5]. We changed “Under pathological conditions, inflammatory bone resorption is associated with many types of cytokines, including tumor necrosis factor-alpha (TNF-a), interleukin 1 (IL-1), and IL-6. IL-1 and IL-6 both stimulate the expression of the major osteoclastogenic factor, receptor activator of nuclear factor kappa-B ligand (RANKL), and TNF-a may induce osteoclasts and is regarded as another osteoclastogenic factor[3,4].” in the introduction section of the original version (page 1, line 40 to page 2, line 1) to “Under pathological conditions, inflammatory bone resorption is associated with many types of cytokines, including tumor necrosis factor-alpha (TNF-a), interleukin 1 (IL-1), and IL-6. IL-1 and IL-6 both stimulate the expression of the major osteoclastogenic factor, receptor activator of nuclear factor kappa-B ligand (RANKL). TNF-a may induce osteoclasts and is regarded as another osteoclastogenic factor [3,4], but anti-bone resorptive effects of TNF-a have been considered through control of inflammation, and anti-TNF-a agents do not have any beneficial effect on interference of bone resorption [5].” in the introduction section of the revised version (page 1, line 40 to page 2, line 3).
Other changes made
We corrected some typographical and grammatical errors using an English Editing Service provided by MDPI.
Round 2
Reviewer 1 Report
The authors addressed my concerns.